# Predictors and experiences of seeking abortion services from pharmacies in Nepal

Leila Harrison[1], Mahesh Puri[2], Diana Greene Foster[3], Sunita Karkia[2], Nadia G. Diamond-Smith[4]*

1 Centre for Global Child Health, The Hospital for Sick Children, Toronto, Canada, 2 Center for Research on Environment, Health and Population Activities (CREHPA), Kathmandu, Nepal, 3 Department of Obstetrics, Gynecology & Reproductive Sciences, University of California, San Francisco, San Francisco, California, United States of America, 4 Department of Epidemiology and Biostatistics, University of California, San Francisco, San Francisco, California, United States of America

* nadia.diamond-smith@ucsf.edu

**Data Availability Statement:** Data relevant to this study are available from the Harvard Dataverse at

## Abstract

Abortion was legalized in Nepal in 2002; however, despite evidence of safety and quality provision of medical abortion (MA) pills by pharmacies in Nepal and elsewhere, it is still not legal for pharmacists to provide medication abortion in Nepal. However, pharmacies often do provide MA, but little is known about who seeks abortions from pharmacies and their experiences and outcomes. The purpose of this study is to understand the experiences of women seeking MA from a pharmacy, abortion complications experienced, and predictors for denial of MA. Data was collected from women seeking MA from four pharmacies in two districts of Nepal in 2021–2022. Data was collected at baseline (N = 153) and 6 weeks later (N = 138). Using descriptive results and multi-variable regression models, we explore differences between women who received and did not receive MA and predictors of denial of services. Most women requesting such pills received MA (78%), with those who were denied most commonly reporting denial due to the provider saying they were too far along. There were few socio-demographic differences between groups, with the exception of education and gestational age. Women reported receiving information on how to take pills and what to do about side effects. Just under half (45%) of women who took pills reported no adverse symptoms after taking them and only 13% sought care. Most women seeking MA from pharmacists in Nepal are receiving services, information, and having few post-abortion symptoms. This study expands the previous limited research on pharmacy provision of MA in Nepal using a unique dataset that recruits women at the time of abortion seeking and follows them over time, overcoming potential biases present in other study designs. This suggests that expansion of the law to allow pharmacy distribution would increase accessibility and reflect current practice.

## Introduction

The legalization of abortion in Nepal was a crucial and important step to improving reproductive health services in the country and reducing the rate of death from abortion-related

DOI: 10.7910/DVN/BBCXI4 (https://doi.org/10.7910/DVN/BBCXI4).

**Funding:** This work was supported by the Packard Foundation (#2019-68460 to DF). The funders had no role in study design, data collection and analysis, decision to publish, or preparation of the manuscript.

**Competing interests:** The authors have declared that no competing interests exist.

complications [1]. Prior to 2002, abortion was not allowed under any circumstances and could be sanctioned by imprisonment. Abortion was legalized in 2002 in Nepal to address the problem of persistently high rates of maternal mortality and morbidity from unsafe abortion in Nepal. The right to access safe abortion services and post-abortion care are fundamental to a women's ability to control their fertility, protect their health, and ensure the wellbeing of their families [2]. Abortion services were offered beginning in 2004 at one hospital in Kathmandu and expanded gradually across the country. In 2018, the government passed the Safe Motherhood and Reproductive Health Rights Act, which made abortion allowable up to 12 weeks' gestation for any reason [1]. It is permitted up to 28 weeks' gestation in cases of rape, incest or in the following situations: if the pregnant women is living with HIV or some other incurable diseases; if the pregnancy poses a danger to the women's life, physical health or mental health; or if there is fetal anomaly. Medication abortion pills (MA) can be legally provided up to 10 weeks' gestation by an auxiliary nurse midwife, staff nurse, midwife, or physician. Above 10 weeks only trained physicians can perform abortions, using manual vacuum aspiration, medical induction and/or dilation and evacuation. Although common, off-label or over the counter sales of MA drugs at pharmacies are not legally permitted in Nepal [3]. As per the policy–both providers and place to obtain abortion services needs to be certified by the government. Pharmacies in Nepal are operated by varied professionals including pharmacists, pharmacy assistants, and other health workers who may have received as little as 48–72 hours orientation on pharmacy services.

Still, significant barriers in accessing safe abortion, such as lack of availability of services, transportation and infrastructure problems, out-of-pocket cost, and lack of knowledge about legal abortions, persist in Nepal [4]. Abortion is reported to be the third leading cause of maternal mortality [5]; and the last national estimate of abortion provision, in 2014, showed that 58% of abortions were performed outside of certified facilities (~186,100 abortions) [1]. Previous study has shown that many people eligible for abortion care are turned away from certified clinics and subsequently seek care elsewhere, including potentially outside of the legal health system [1, 2, 4]. Many people who seek abortion in clinics have previously attempted to obtain abortion pills at pharmacies, even though pharmacists may not be trained and are not legally certified to dispense the pills for this indication [2]. Pharmacies tend to be more accessible than health facilities and are therefore often the first point of contact for women seeking abortion information [4].

Given access to safe care remains constrained in Nepal, one approach to improve abortion access has been to improve the availability of MA (i.e., mifepristone and misoprostol tablets) during the first trimester [6]. MA typically does not require laboratory tests, ultrasound, or sterile equipment, which are particularly challenging to access in rural low-resource settings. The Nepal Demographic and Health Survey (2016) reported that MA accounted for 70% of all abortions in Nepal, and 19% of these MA were obtained from a pharmacy [3, 7]. Though they lack legal authority to prescribe and dispense, many pharmacies in Nepal stock MA [1]. Mifepristone-misoprostol abortion has been shown to be as effective and safe when provided by trained auxiliary nurse-midwives at pharmacies as at government-certified health facilities. Nevertheless, pharmacy prescribing and dispensing of abortion pills is not legally permitted in Nepal, there is no widespread training of pharmacists about abortion, and there is little known about where, how, and from whom women obtain abortion pills at pharmacies. As such, there is a gap in understanding what information people seeking abortion receive at pharmacies, whether people are screened appropriately, and whether people get appropriate follow up care.

Ensuring that women seeking abortion services from pharmacies receive safe and effective care continues to be a challenge in Nepal [1]. Conflicting evidence has been reported in the literature for provision of pharmacy assisted abortions. A systematic review of studies from

around the world (including Nepal) show a wide variation in MA practices and poor knowledge among pharmacists, where untrained pharmacy workers often provide inadequate or inaccurate information and may dispense unsafe or ineffective MA pills [8]. However, a more recent systematic review restricted to studies on MA in Nepal found that trained pharmacy workers can safely and effectively provide MA [1].

Although studies have been conducted in other countries, gaps remain in our understanding of women's experiences with seeking abortion pills at pharmacies in Nepal. Further, existing studies are of a qualitative nature, thus requiring additional quantitative research to substantiate the findings. We know from extensive research from the United States Turnaway study [9], that women who seek an abortion but are denied, have significant negative consequences to socioeconomic status, physical and mental health, and wellbeing, when compared to women who received the abortion. In addition to needing quantitative information about who seeks care in pharmacies and women's experiences receiving MA from pharmacies in Nepal, we do not know if women are receiving MA from pharmacies and what factors are associated with being denied. We know from our work recruiting women seeking abortion from certified clinics [3], that almost one in ten had previously sought abortion care from pharmacies and either did not get the wanted MA pills or the pills they received were not effective in ending their pregnancy, thereby necessitating a trip to a clinic. In this paper, we present data from a longitudinal study among women who sought MA from pharmacies in Nepal. The study aimed to understand the experiences of women seeking MA from pharmacies, abortion complications experienced, and predictors of denial of MA.

## Materials and methods

This study was conducted at four pharmacy locations in two districts (Chitwan and Rupendehi) in Nepal. These two districts were selected purposively to have diverse representation, availability of abortion data, available list of pharmacies from the previous study, built in trust form previous work and willingness to support the study. These four pharmacies, selected from a list of 19 compiled by the Center for Research on the Environment, Health, and Population Activities (CREHPA), showed a sufficient number of women seeking pregnancy termination services who were willing to participate. Additional eligibility criteria included: 1) pharmacy reported that they sell any medication abortion drugs (registered or unregistered) to their clients, 2) there are separate locked cabinets or storage closets for study data; and 3) there is a private room for conducting interviews with the participants.

The Nepal Turnaway Study is a 4-year prospective cohort study examining the effects of unwanted pregnancy on women's lives in Nepal. Study design and methods have been published previously [10]. Most data were collected from women seeking abortions at certified clinics; however, a small sub-study was conducted among women seeking abortions at pharmacies. The present study analyzes the longitudinal data collected for the Nepal Turnaway Study, focusing on a subgroup of the sample who sought abortion care from pharmacies.

The sample included women who were over 15 years, seeking abortion care from a pharmacy, and living in Nepal. We enrolled women seeking abortion services from the pharmacies between 15 March 2021 and 31 March 2022. The sample size was based on budget and time restrictions, and because it was uncertain if women would be willing to participate given the sensitive nature of receiving medication abortion illegally from a pharmacist, it was not powered to detect differences between groups (clinic/pharmacy; denied/received).

All women seeking termination of pregnancy were screened for study eligibility by the pharmacist or any person working at the pharmacy. The pharmacist completed an eligibility form for every woman seeking an abortion over the enrollment period. The form recorded the

woman's age, marital status, number of living children, estimated gestation, provider assessment of eligibility for abortion, and reason for ineligibility, if applicable. If a woman was eligible for the study, the pharmacist referred her to speak with a trained research staff member who was stationed in a private room at each pharmacy. The research staff member confirmed study eligibility, obtained written informed consent (for women unable to sign, a thumbprint was obtained), and conducted the baseline survey in the pharmacy using a tablet, which saved survey responses to a secure web-based storage platform. Interviewers then contacted participants six weeks later with the plan to do surveys every six months for the next three years. The interviewer conducted the surveys in Nepali. Each participant received financial compensation equivalent to about 4 USD for the baseline and each subsequent interview.

Ethics Statement: This study was approved by the Institutional Review Board of the University of California, San Francisco and the Nepal Health Research Council, Nepal. Written informed consent (or for women unable to sign, a thumbprint) was obtained for all participants. In Nepal those under age 16 are considered to be minors and those who are married are considered able to provide consent,: all participants in this study were 17 or older.

Our primary outcome of interest was whether the participant received or was denied an abortion from a pharmacist. Additionally, a number of sociodemographic variables were included in this analysis based on previous studies in this setting [1, 3, 5, 6]: age (years), marital status (married, never married, divorced/separated, or widowed), caste (Brahmin, Chhetri, Thakuri, Hill Janajati, Terai Janajati, Dalit, other specified), highest level of formal and non-formal schooling attained (no schooling, some non-formal, primary, secondary, or higher), employment status (yes/no), and number of children (0, 1, 2, 3 or more). In addition, we created an objective measure of socioeconomic status using asset variables to determine each participants' wealth quintile, by creating a score from a set of assets validated in Nepal previously [8]. The wealth quintile variable was created using principal component analysis on wealth factors (e.g., access to drinking water, electricity, type of toilet facility, type of cooking fuel, type of house, materials used for roof and wall, owning a television, computer, fridge) and created quintiles based on the distribution of these factors.

To determine the experiences of women who received MA pills from the pharmacy, we included two variables: 1) symptoms or complications following abortion and 2) ability to continue with daily tasks following the abortion. Specifically, we asked participants, 'Did you have any of the following symptoms or complications since having your abortion?' and respondents could select all that applied from a list of possible symptoms and complications. The ability to continue with daily tasks following abortion was assessed with the question, 'Was there a period after your abortion when you were physically unable to walk, climb steps, or do chores?'. Both variables were collected at follow-up survey time points.

First, we used descriptive statistics to describe the sociodemographic differences between women who did and did not receive an abortion from a pharmacy. Differences between groups were determined through bivariate analysis using fisher's exact or chi2 tests where appropriate. We then conducted multivariate logistic regression to determine socio-demographic factors associated with being denied an abortion at pharmacies. All analyses were conducted using STATA 17.0.

## Results

Between 15 March 2021 and 31 March 2022, 162 women presented for abortion at the four selected pharmacies. A total of 153 women (94%) consented to participate and completed the baseline interviews in this study and 138 completed the 6 weeks survey. Of these 138 women, 107 received MA pills and 31 were denied. In the context of a legal limit of 10 weeks at clinics,

**Table 1. Baseline sociodemographic characteristics of women who received compared to were denied a pharmacy abortion (N = 138).**

| Variable | Received medication abortion pills (N = 107) | Denied medication abortion pills (N = 31) | Test statistic and p-value |
|---|---|---|---|
| *Age (yrs)* | | | |
| 17–19 (n = 6) | 4 (67%) | 2 (33%) | Fisher's exact p-value = 0.161 |
| 20–35 (n = 104) | 78 (75%) | 26 (25%) | |
| 36–45 (n = 28) | 25 (89%) | 3 (11%) | |
| *Marital status* | | | |
| Ever married (n = 128) | 101 (79%) | 27 (21%) | Fisher's exact p-value = 0.23 |
| Never married (n = 10) | 6 (60%) | 4 (40%) | |
| *Caste* | | | |
| Brahmin/Chhetri/Thakuri (n = 55) | 43 (78%) | 12 (22%) | Fisher's exact p-value = 0.74 |
| Hill Janajati/Terai Janajati (n = 61) | 47 (77%) | 14 (23%) | |
| Dalit (n = 17) | 12 (71%) | 5 (29%) | |
| Other (n = 5) | 5 (100%) | 0 (0%) | |
| *Education* | | | |
| No schooling (n = 8) | 4 (50%) | 4 (50%) | Fisher's exact p-value = 0.026 |
| Some non-formal or primary (n = 19) | 12 (63%) | 7 (37%) | |
| Secondary or higher (n = 111) | 91 (82%) | 20 (18%) | |
| *Wealth quintile* | | | |
| 1 (n = 28) | 19 (68%) | 9 (32%) | |
| 2 (n = 28) | 20 (71%) | 8 (29%) | Fisher's exact p-value = 0.19 |
| 3 (n = 28) | 23 (82%) | 5 (18%) | |
| 4 (n = 28) | 21 (75%) | 7 (25%) | |
| 5 (n = 26) | 24 (92%) | 2 (8%) | |
| *Employment* | | | |
| Are you currently working (not including housework)? | | | |
| Yes (n = 75) | 60 (80%) | 15 (20%) | $X^2(1) = 0.57$, p = 0.45 |
| No (n = 63) | 47 (75%) | 16 (25%) | |
| *Number of children* | | | |
| 0 (n = 19) | 15 (79%) | 4 (21%) | Fisher's exact p-value = 1.00 |
| 1 (n = 40) | 31 (78%) | 9 (22%) | |
| 2 (n = 56) | 43 (77%) | 13 (23%) | |
| 3 or more (n = 23) | 18 (78%) | 5 (22%) | |
| *Gestational age at baseline* | | | |
| *<10 weeks (n = 112)* | 107 (96%) | 5 (4%) | Fisher's exact p-value = 0.00 |
| *≥10 weeks (n = 26)* | 0 (0%) | 26 (100%) | |

it is important to note that 112 women were <10 weeks, of whom 96% received their abortion; 26 women were over 10 weeks, none of whom received MA from the pharmacy (Table 1).

Most (N = 112, 81%) reported time since last period of less than 10 weeks, making them eligible to receive MA from a certified clinic. Most (N = 107, 78%) received MA from the pharmacy and 22% (N = 31) were denied. The most common reason that women were denied was that the pharmacist said they were too far along (81%) (Table 2). Other responses included not having enough money, not knowing their date of last menstrual period, other medical problems, and no reason given.

From bivariate analyses, there were few significant differences in sociodemographic characteristics between women who received MA pills at the pharmacy and those denied (Table 1).

**Table 2. Reasons for denial.**

| Reason (N = 31) | n (%) |
|---|---|
| Provider/pharmacist said I was too far along | 25 (81%) |
| I was not sure I wanted an abortion | 2 (6%) |
| Didn't have money | 1 (3%) |
| Provider/pharmacist did not give a reason | 1 (3%) |
| Provider/pharmacist said I have other medical problems so they couldn't do the abortion | 1 (3%) |
| Pharmacist asked about my last menstrual period and I did not know | 1 (3%) |
| I could afford but didn't want to pay high amount they said. I knew other pharmacy that sells pills much cheaper than here. | 1 (3%) |

The age distribution was similar across both groups, with most women between the ages of 20–35 years. For both groups, most women were married, had two children or more, were predominantly of castes Brahmin/Chhetri/Thakuri and Hill Janajati/Terai Janajati, and had secondary or higher levels of education. Gestational age was highly associated with abortion denial: no women who reported being more than 10 weeks pregnant received abortion pills from the pharmacy.

Women could report more than one reason for seeking care at the pharmacy that they choose, and the most common reasons were it being the easiest to get to (54%), closest to their home (36%), had a good reputation (29%) or it was recommended by a friend (22%) (Table 3).

Nearly all women reported getting information from the pharmacist about how to take the medication abortion pills (99%) (Table 4). Slightly fewer women received information about symptoms to expect after taking pills (90%) and when they should seek care if experiencing symptoms (83%).

For women who received MA pills at the pharmacy, 45% did not report having any symptoms (Table 5) after taking MA pills. For those who did, the most frequently reported were abdominal pain (36%) and heavy bleeding more than expected (33%). About a quarter (24%) reported they were unable to walk, climb steps, or do chores following their abortion. Less than a sixth (13%) reported seeking treatment for their symptoms. Open-ended responses (not shown in the table) indicated that of the 15 women who sought care, three received procedures for incomplete abortion, four received medicine to control their perceived ongoing heavy

**Table 3. Reasons for choosing pharmacy for abortion services (N = 138*).**

| Why did you choose to come to this pharmacy? | n (%) |
|---|---|
| It was the easiest to get to | 75 (54%) |
| It was the closest to my house | 50 (36%) |
| It has a good reputation | 40 (29%) |
| It was recommended by friends/family | 31 (22%) |
| It provides confidential service | 7 (5%) |
| It was the cheapest | 6 (4%) |
| It was recommended by a health care provider | 6 (4%) |
| It was recommended by my husband | 6 (4%) |
| Other | 5 (3%) |

*Note: Some women reported multiple responses

**Table 4. Information women received from pharmacy among those who received medication abortion pills (N = 107).**

| Question asked | Response | n (%) |
|---|---|---|
| Did you get information from the pharmacist about how to take the pills? | No | 1 (1%) |
| | Yes | 106 (99%) |
| Did you get information from the pharmacist about what symptoms to expect? | No | 11 (10%) |
| | Yes | 96 (90%) |
| Did you get information from the pharmacist about what symptoms mean you should seek care at a health clinic? | No | 18 (17%) |
| | Yes | 89 (83%) |

bleeding, and the remaining seven received other types of treatment/counseling but no other medications or procedure. No one reported that they knew they were still pregnant at 6 weeks, although two were not sure.

In multivariable logistic regression analyses, for each additional year of age, women had lower odds of being denied their abortion (OR = 0.85, 95% CI: 0.72–0.99) (Table 6). Women with any schooling–non-formal or primary school–compared to no schooling, had significantly lower odds of being denied an abortion (OR = 0.04, 95% CI: 0.00–0.60). Similarly, women with secondary school or higher education, compared to no schooling, had significantly lower odds of being denied an abortion (OR = 0.03, 95% CI: 0.00–0.29). Women who were ≥10 weeks gestational age or unsure of their gestational age at baseline, compared to <10 weeks, had much higher odds of being denied (OR = 121.63, 95% CI: 23.17–638.50).

## Discussion

Most women in our study–and most who were under 10 weeks' gestation—who sought abortion care from pharmacies in Nepal received MA services. This suggests that although the

**Table 5. Symptoms and care-seeking after a medication abortion (N = 106$*).**

| Symptoms/care-seeking | n (%) |
|---|---|
| No symptoms | 48 (45%) |
| Abdominal pain | 38 (36%) |
| Heavy bleeding | 35 (33%) |
| Vomiting | 11 (10%) |
| Nausea | 9 (8%) |
| Chills/shivering | 8 (8%) |
| Fever | 7 (7%) |
| Fainting | 3 (3%) |
| *Physically unable to walk, climb steps, or do chores following abortion* | |
| Yes | 25 (24%) |
| No | 81 (76%) |
| *Sought care from health care provider due to symptoms* | 14 (13%) |
| *Still pregnant at 6 weeks* | |
| *Yes* | 0 (0%) |
| *Don't know* | 2 (2%) |
| *No* | 104 (98%) |

$Sample drops to 106 because 1 person out of the total 107 who received MA said they did not take pills

*Note: Some women reported experiencing multiple symptoms

**Table 6. Predictors of being denied an abortion from pharmacy (N = 133).**

| Variable | OR (95% CI) | p-value |
|---|---|---|
| Age (continuous) | **0.85 (0.72, 0.99)** | **0.04** |
| Married | | |
| No | REF | - |
| Yes | 2.61 (0.27, 25.49) | 0.41 |
| Caste | | |
| Brahmin/Chhetri/Thakuri | REF | - |
| Hill Janajati | 0.57 (0.13, 2.60) | 0.47 |
| Dalit | 1.04 (0.14, 7.70) | 0.97 |
| Terai Janajati | 2.53 (0.24, 26.87) | 0.77 |
| Education | | |
| No schooling | REF | - |
| Some non-formal or primary school | **0.04 (0.00, 0.60)** | **0.02** |
| Secondary or higher | **0.03 (0.00, 0.29)** | **0.00** |
| Wealth quintile (continuous) | 1.16 (0.67, 2.02) | 0.60 |
| Employment | | |
| No | REF | - |
| Yes | 0.80 (0.21, 3.14) | 0.75 |
| Number of children (continuous) | 2.49 (0.83, 7.51) | 0.11 |
| Gestational age at baseline | | |
| <10 weeks | REF | - |
| ≥10 weeks or don't know | **121.63 (23.17, 638.50)** | **0.00** |

*Bold = significant

provision of MA is not legally permitted through pharmacies, pharmacists who are providing MA are doing so according to the legal guidelines regarding gestational age (10 weeks or less gestational age). In contrast, a recent study using mystery clients (both male and female) found that pharmacies in Nepal dispensed MA pills in response to only just over a third of requests [11]. The study reports gender discrepancies in behaviour of pharmacy workers, thus the use of both male and female clients may explain one reason why the earlier study found a higher level or refusal. It is possible that had this study restricted their sample to women reporting that their pregnancy was under 10 weeks' gestation, the proportion of clients who received MA services would have been higher and more similar to our study findings.

The main predictor of receipt of abortion in our study was gestational age at the time of abortion seeking; thus, other factors associated with denial may be more associated with advanced gestational age, rather than denial of care per se. Younger women and those with lower education may be less likely to recognize the symptoms of pregnancy and therefore later in seeking care. More work should be done to determine whether young age or low education are independent predictors of denial of MA at pharmacies. Other sociodemographic factors such as marital status, wealth or other factors that typically act as barriers or lead to provider bias do not seem to be impeding MA access in pharmacies in Nepal. Educational status is likely associated with women being more able to advocate for services. Previous research in India found that women received differential care at pharmacies by age and marital status, suggesting provider bias [12]. Fortunately, this does not seem to be occurring in these pharmacies in Nepal. Our sample was small and thus a larger study in a more diverse population might find associations between socio-demographics and abortion receipt.

Not only are women receiving MA pills through pharmacies, but most are also getting information from the pharmacy workers about what to expect. Perhaps partly because of this, women are not experiencing many symptoms that concern them, and few need to seek care elsewhere. Only three of the 111 women who received MA services in our sample had an incomplete abortion. A larger recent study from Nepal found a similarly small percentage (21 out of 992, 2%) experienced an incomplete abortion after taking MA pills obtained from a pharmacy [13]. Symptoms reported by women were similar to what other studies have suggested is normal in the progression of a medication abortion [14–19].

Although pharmacy provision of medication abortion pills is not formally permitted under the Nepal Safe Motherhood and Reproductive Health Rights Act of 2018, many people seek abortion pills through pharmacies and may only seek clinical care when they are unable to get such pills or experience an incomplete abortion [3]. Our small study of women who sought abortion pills directly from a pharmacy shows that pharmacists are capable of complying with the rules around gestational age screening and that many also provide critical information about what to expect in terms of symptoms and when to seek follow up care.

This study has strengths and limitations. Its value is its unique approach in recruiting women at the time of abortion seeking, and following them over time, thus eliminating some potential biases due to recall and sensitivities reporting abortion seeking or receipt. Limitations include the small sample size and fact that we recruited from only four pharmacies, in only two districts of Nepal, which were mostly urban, however these did represent some geographic heterogeneity across the country. Data was collected during the COVID-19 pandemic, thereby participants and their experiences may have differed from those seeking MA at other times. We could hypothesize that COVID-19 would have acted as a barrier to seeking care or led to more women having unintended pregnancies due to gaps in reproductive care access [20, 21]. Despite these limitations, this study provides insight into the experiences of women seeking abortions at pharmacies in Nepal, an understudied but likely common experience.

## Conclusions

Although pharmacies not currently registered as legal providers of abortion pills, many women seek medication abortion from pharmacies in Nepal. We find that women who are below the recommended clinical gestational age for medication abortion pills are able to obtain pills at pharmacies along with information in Nepal. Many women who seek abortion services through pharmacies do so because such services are easy to access and pharmacies are sites recommended by family and friends. Expansion of the law to allow pharmacy distribution would increase accessibility for people who cannot easily access government certified facilities. A change in law would reflect current practice in the country whereby pharmacies are a common first stop in seeking healthcare services.

## Supporting information

**S1 Checklist. Inclusivity in global research.**
(DOCX)

## Acknowledgments

We would like to thank Dev Chandra Maharjan and Sarah Raifman, as well as the women who gave their time to participate in the study.

## Author Contributions

**Conceptualization:** Diana Greene Foster.

**Formal analysis:** Leila Harrison, Nadia G. Diamond-Smith.

**Funding acquisition:** Diana Greene Foster.

**Investigation:** Mahesh Puri, Sunita Karkia.

**Methodology:** Mahesh Puri, Diana Greene Foster, Nadia G. Diamond-Smith.

**Project administration:** Sunita Karkia.

**Supervision:** Mahesh Puri, Nadia G. Diamond-Smith.

**Writing – original draft:** Leila Harrison.

**Writing – review & editing:** Mahesh Puri, Diana Greene Foster, Nadia G. Diamond-Smith.

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
