## [Decision Letter · Decision Letter 0]

16 Jan 2024

PGPH-D-23-02278

Predictors and experiences with pharmacy abortion denial in Nepal

Dear Dr. Diamond-Smith,

Thank you for submitting your manuscript to PLOS Global Public Health. After careful consideration, we feel that it has merit but does not fully meet PLOS Global Public Health’s publication criteria as it currently stands. Therefore, we invite you to submit a revised version of the manuscript that addresses the points raised during the review process.

Editor comments:

The reviewers and I found the manuscript to be well-written and timely, offering important insights into women's abortion seeking at pharmacies in Nepal, as well as predictors of abortion receipt.However, the reviewers note several areas where additional information and clarification of methods and results would strengthen the manuscript. In particular, the selection of the districts and pharmacies included, and the context of these districts would be very helpful for readers. I would also encourage the authors to include their survey instrument, along with scale characteristics, as a supplemental file.

We look forward to receiving your revised manuscript.

Kind regards,

Marie A. Brault, PhD

Academic Editor

Journal Requirements:

1. Please include the following request in the decision letter, and ping me with follow up. “Please include a complete copy of PLOS’ questionnaire on inclusivity in global research in your revised manuscript. Our policy for research in this area aims to improve transparency in the reporting of research performed outside of researchers’ own country or community. The policy applies to researchers who have travelled to a different country to conduct research, research with Indigenous populations or their lands, and research on cultural artefacts. The questionnaire can also be requested at the journal’s discretion for any other submissions, even if these conditions are not met.  Please find more information on the policy and a link to download a blank copy of the questionnaire here: https://journals.plos.org/globalpublichealth/s/best-practices-in-research-reporting. Please upload a completed version of your questionnaire as Supporting Information when you resubmit your manuscript.

2. We noticed that you used "not shown" in the manuscript. We do not allow these references, as the PLOS data access policy requires that all data be either published with the manuscript or made available in a publicly accessible database. Please amend the supplementary material to include the referenced data or remove the references.

Additional Editor Comments (if provided):

Reviewers' comments:

Reviewer's Responses to Questions

**Comments to the Author**

1. Does this manuscript meet PLOS Global Public Health’s publication criteria? Is the manuscript technically sound, and do the data support the conclusions? The manuscript must describe methodologically and ethically rigorous research with conclusions that are appropriately drawn based on the data presented.

Reviewer #1: Partly

Reviewer #2: Partly

2. Has the statistical analysis been performed appropriately and rigorously?

Reviewer #1: Yes

Reviewer #2: I don't know

3. Have the authors made all data underlying the findings in their manuscript fully available (please refer to the Data Availability Statement at the start of the manuscript PDF file)?

Reviewer #1: Yes

Reviewer #2: No

4. Is the manuscript presented in an intelligible fashion and written in standard English?

Reviewer #1: Yes

Reviewer #2: Yes

5. Review Comments to the Author

Reviewer #1: 01) Current needs to be improved. The term "pharmacy abortion denial" might be rephrased for smoother readability. Consider using terms like "pharmacy-assisted abortion denial" for precision and coherence. While the current title focuses on denial, the positive outcomes and experiences of those who successfully receive pharmacy-assisted abortion services are not emphasized. Including a mention of both denial predictors and positive experiences could provide a more balanced perspective.

Ex:- "Beyond Legal Boundaries: Exploring Pharmacy-Assisted Abortion in Nepal and Predictors of Denial"

"Pharmacists as Gatekeepers: Examining Experiences and Denial Predictors of Abortion Services in Nepal"

02) Line 142 - Could the authors provide clarity on the rationale behind selecting Chitwan and Rupendehi districts for the study? Was the selection based on purposive sampling to ensure a diverse representation, or was it more of a convenience sampling approach? Elaborating on the criteria for district selection would enhance the reader's understanding of the study's context and strengthen the methodological transparency.

03) "On line 158, the authors specify that the study included women aged 17-45 years. Given that the typical reproductive lifespan of women extends up to 49 years, it would be beneficial for the authors to provide a rationale for the chosen age range. Clarifying the reasons behind this decision would help readers understand the scope of the study and any potential implications for generalizability to the broader reproductive-age population."

04) On line 191, the authors state that they created an objective measure of socioeconomic status using asset variables to determine participants' wealth quintile. To ensure the validity and reliability of the measurement scale, could the authors provide information on whether a standardized measurement scale, validated in the context of Nepal, was employed? If so, including references to the validated measurement scale would strengthen the methodological rigor and contextual relevance of the socioeconomic status assessment in this study.

05) On line 208, the authors mention that differences between groups were determined using bivariate analysis and t tests. Considering the potential impact of non-normality on the validity of t tests, it would be valuable for the authors to clarify whether normality assumptions were tested. If the data did not meet the normality assumption, a mention of the use of nonparametric tests for rank data, which are more suitable in such cases, would enhance the robustness of the statistical analysis.

06) I could not locate information on the sample size and the procedure followed for sample size calculation in the manuscript. To ensure transparency and interpretability of the study findings, it is crucial to include details regarding the rationale behind the chosen sample size and the method used for its determination. Please provide this information in the manuscript to enhance the overall methodological clarity and robustness of the study.

07) In Table 1, the section on 'Reasons for denial' appears to have limited variability, with only one respondent categorizing for most options. This raises concerns about the informativeness of the data in this table. It may be beneficial to reconsider the presentation of this information in other form.

08) In Table 2, it would be valuable to enhance the presentation of the baseline sociodemographic characteristics by adding an additional column to include p-values and the corresponding test statistics. This addition would offer readers a clear understanding of the type of test used and the statistical significance of differences between women who received and were denied pharmacy abortion services.

09) In Table 3, the section on 'Reasons for choosing pharmacy for abortion services' could be simplicity. Considering the limited frequency (one or two) for the last four categories, it may be beneficial to merge them into a single category labeled 'Other.'

10) To improve the readability of Table 6, it is suggested to merge the second column (Odds Ratio) and the third column (95% Confidence Interval of the Odds Ratio). This can be achieved by presenting the Odds Ratio and its corresponding confidence interval in a single column as 'OR (95% CI),' for example: '0.85 (0.72-0.99)' for age.

11) While the study provides valuable insights, it would be beneficial to include a dedicated section discussing the limitations. Consider addressing factors such as the sample size, the number of pharmacies selected, and the adopted methodology.

12) In addition to the valuable insights gained from this study, it is recommended to incorporate a dedicated 'Conclusions' section. This section can serve as a crucial endpoint to succinctly summarize and discuss the main findings, drawing connections to the broader implications for policy, practice, and future research.

Reviewer #2: Review for “Predictors and experiences with pharmacy abortion denial in Nepal”

I appreciated the opportunity to review this manuscript regarding data from a longitudinal study of women’s experiences seeking and using medical abortion pills from pharmacies in Nepal. The topic and findings are very timely, given the political climate surrounding abortion in Nepal and around the world. I have a few comments and questions to help clarify and strengthen the manuscript.

General Comments

• This manuscript discusses pharmacy provision of abortion services. Does this mean provision without a prescription, with a prescription, or either? It would be helpful if you were more direct in stating that the study concerns illegal, over-the-counter abortion pill provision, if that is the case. Particularly in the abstract, the language about that is vague.

• The manuscript discusses biases influencing who is approved or denied the abortion medication, but the results suggest that the decision is based almost entirely on whether participants meet the 10-week guideline, indicating that there is very little bias occurring. An analysis of women who are eligible based on the 10-week guideline, but denied anyway, would be necessary to make assertions about biases. The current study only has 5 individuals out of 143 who meet that description.

Introduction

• It would be helpful to see a clearer explanation regarding the legality/process of obtaining abortion pills from the pharmacy (Is it legal with a prescription from a physician/midwife? How often do pharmacies employ an auxiliary nurse-midwife? Do the women or the pharmacists face legal consequences if caught? If so, what are they?)

• Lines 117-122: If the word limit permits, it would be useful to provide more information about the conflicting Nepal studies regarding pharmacy assisted abortion, beyond the fact that they conflict (i.e., what were the methods/samples and specific findings?)

Materials and Methods

• Were there only four eligible pharmacies or did you select four from a greater number?

• I would be interested to know what steps were taken to protect pharmacists and women from legal repercussions for participating.

Results

• The logistic regression analysis is misleading, because it doesn’t show what it purports to show regarding biases.

o According to guidelines, the cut off for abortion pill access is 10 weeks pregnant. Based on the data from table 2, whether a woman meets this guideline appears to be, by far, the main factor in whether she is approved or denied the medication (100% of approved women are under 10 weeks pregnant, 84% of denied women were over 10 weeks pregnant, and 96% of women under 10 weeks were approved).

o In effect, this regression is basically a comparison of people under 10 weeks and people over 10 weeks pregnant at the time abortion is sought (with a few exceptions).

o If you wanted to do a logistic regression to look at biases among who is approved or denied, you would need to look at people who are eligible based on the 10-week guidelines only, and compare differences in who is approved vs who is denied among that group. Of course, that’s not possible with only a handful of participants under 10 weeks being denied.

Discussion

• Minor, but lines 287-288 state that “All who were under 10 weeks’ gestation” received medical abortion services, but it was actually 96% according to Table 2.

6. PLOS authors have the option to publish the peer review history of their article (what does this mean?). If published, this will include your full peer review and any attached files.

**Do you want your identity to be public for this peer review?** For information about this choice, including consent withdrawal, please see our Privacy Policy.

Reviewer #1: <

---

## [Decision Letter · Decision Letter 1]

8 Apr 2024

Predictors and experiences of seeking abortion services from pharmacies in Nepal

PGPH-D-23-02278R1

Dear Dr. Diamond-Smith,

We are pleased to inform you that your manuscript 'Predictors and experiences of seeking abortion services from pharmacies in Nepal' has been provisionally accepted for publication in PLOS Global Public Health.

Best regards,

Marie A. Brault, PhD

Academic Editor

Reviewer Comments (if any, and for reference):

Reviewer's Responses to Questions

**Comments to the Author**

1. If the authors have adequately addressed your comments raised in a previous round of review and you feel that this manuscript is now acceptable for publication, you may indicate that here to bypass the “Comments to the Author” section, enter your conflict of interest statement in the “Confidential to Editor” section, and submit your "Accept" recommendation.

Reviewer #1: All comments have been addressed

Reviewer #2: All comments have been addressed

2. Does this manuscript meet PLOS Global Public Health’s publication criteria? Is the manuscript technically sound, and do the data support the conclusions? The manuscript must describe methodologically and ethically rigorous research with conclusions that are appropriately drawn based on the data presented.

Reviewer #1: Yes

Reviewer #2: Yes

3. Has the statistical analysis been performed appropriately and rigorously?

Reviewer #1: Yes

Reviewer #2: Yes

4. Have the authors made all data underlying the findings in their manuscript fully available (please refer to the Data Availability Statement at the start of the manuscript PDF file)?

Reviewer #1: Yes

Reviewer #2: No

5. Is the manuscript presented in an intelligible fashion and written in standard English?

Reviewer #1: Yes

Reviewer #2: No

6. Review Comments to the Author

Reviewer #1: Congratulations for the successful research. All the best with future works.

Reviewer #2: I appreciated the opportunity to re-review this manuscript regarding pharmacy-assisted abortion in Nepal. The authors adequately addressed all of my suggestions. I particularly appreciated the updated language regarding biases (or lack thereof) in refusal of care.

To strengthen the manuscript, I would recommend additional copyediting for grammar and clarity. For example, please see some of the minor grammatical errors listed below:

Lines 46-47: “However, pharmacies often do provide MA, but little is known about who seeks abortions from pharmacies and their experiences and outcomes.” [run-on sentence]

Line 75-76: “the right to…are fundamental” [subject-verb agreement]

Line 82: “a women’s ability”

Line 88: “As per the policy –“ [use a comma rather than a dash]

Line 96: “Abortion is reported to be the third leading cause of maternal mortality; and the last national estimate of abortion provision” [ “and” should not follow a semicolon]

7. PLOS authors have the option to publish the peer review history of their article (what does this mean?). If published, this will include your full peer review and any attached files.

**Do you want your identity to be public for this peer review?** For information about this choice, including consent withdrawal, please see our Privacy Policy.

Reviewer #1: **Yes: **M. Suchira Suranga

Reviewer #2: No
